# New Aspects of Vitamin K Research with Synthetic Ligands: Transcriptional Activity via SXR and Neural Differentiation Activity

**DOI:** 10.3390/ijms20123006

**Published:** 2019-06-20

**Authors:** Yoshihisa Hirota, Yoshitomo Suhara

**Affiliations:** 1Laboratory of Biochemistry, Department of Bioscience and Engineering, College of Systems Engineering and Science, Shibaura Institute of Technology, 307 Fukasaku, Minuma-ku, Saitama 337-8570, Japan; hirotay@shibaura-it.ac.jp; 2QOL Improvement and Life Science Consortium, Shibaura Institute of Technology, 307 Fukasaku, Minuma-ku, Saitama 337-8570, Japan; 3Laboratory of Organic Synthesis and Medicinal Chemistry, Department of Bioscience and Engineering, College of Systems Engineering and Science, Shibaura Institute of Technology, 307 Fukasaku, Minuma-ku, Saitama 337-8570, Japan

**Keywords:** vitamin K, ã-glutamyl carboxylase (GGCX), steroid and xenobiotic receptor (SXR), neural differentiation action, UBIAD1, derivatives research

## Abstract

Vitamin K is classified into three homologs depending on the side-chain structure, with 2-methyl-1,4-naphthoqumone as the basic skeleton. These homologs are vitamin K_1_ (phylloquinone: PK), derived from plants with a phythyl side chain; vitamin K_2_ (menaquinone-*n*: MK-*n*), derived from intestinal bacteria with an isoprene side chain; and vitamin K_3_ (menadione: MD), a synthetic product without a side chain. Vitamin K homologs have physiological effects, including in blood coagulation and in osteogenic activity via γ-glutamyl carboxylase and are used clinically. Recent studies have revealed that vitamin K homologs are converted to MK-4 by the UbiA prenyltransferase domain-containing protein 1 (UBIAD1) in vivo and accumulate in all tissues. Although vitamin K is considered to have important physiological effects, its precise activities and mechanisms largely remain unclear. Recent research on vitamin K has suggested various new roles, such as transcriptional activity as an agonist of steroid and xenobiotic nuclear receptor and differentiation-inducing activity in neural stem cells. In this review, we describe synthetic ligands based on vitamin K and exhibit that the strength of biological activity can be controlled by modification of the side chain part.

## 1. Introduction

Vitamin K plays an important role in blood coagulation and bone formation and has thus been clinically applied for the treatment and prevention of bleeding and osteoporosis. This compound is classified into various homologs that share a basic skeleton of 2-methyl-1,4-naphthoquinone and differ by side-chain structure [1,2]. Natural vitamin K includes phylloquinone (PK), derived from plants (e.g., green vegetables), and menaquinones (MK-*n*), derived from intestinal bacteria and fermented foods (e.g., cheeses and the Japanese soybean product natto) [3,4] (Figure 1). The dominant dietary form of vitamin K in the United States and Europe is phylloquinone (90% of daily intake), whereas the major form in Japan is menaquinones (10%), especially menaquinone 7 (MK-7) [5,6,7]. PK has a phythyl side chain, whereas MK-*n* comprises homologs that vary in the number (*n*) of isoprenyl groups of the side chain (from *n* = 1 to *n* = 14) [8]. Among the vitamin K homologs, MK-4 (*n* = 4) shows the greatest variety of physiological activities [9]. Although PK and MK-4 have the same number of carbon atoms in the side chain, they differ in the degree of unsaturation. In Japan, PK is applied as an antihemorrhagic agent and MK-4 as a therapeutic agent for osteoporosis. MK-4 is present at high concentrations in human, mouse, and rat tissues [10,11,12]. Although humans generally obtain PK and MK-7 from the diet, intake of MK-4 in animal foods is extremely low. Menadione (MD), a synthetic vitamin K analog (Figure 1), is the primary source of vitamin K in rat and mouse feed, along with small amounts of PK. Thus, although humans, mice, and rats ingest only negligible amounts of MK-4, it is the most abundant vitamin K homolog in the tissues. To resolve this apparent inconsistency, we have focused on elucidating the mechanism for converting ingested vitamin K to MK-4. Our work has demonstrated that vitamin K ingested from the diet undergoes a side-chain cleavage reaction in the small intestine to be converted into a form (MD) with no side chain. MD then migrates throughout the body via the lymphatic vessels and is converted to MK-4 by UbiA prenyltransferase domain-containing protein l (UBIAD1), which is present in all tissues, with particularly high levels in the brain [13,14,15,16]. Therefore, we anticipated that MK-4 plays an important functional role, but almost no derivative studies of vitamin K are available. Given this background, we focused on the synthesis of vitamin K derivatives to obtain more potent active compounds. Here, we provide an overview highlighting our findings from this work with synthetic derivatives, with an emphasis on newly identified physiological roles for vitamin K.

## 2. Physiological Effects Identified in Studies of the Vitamin K Cycle and Derivatives Targeting γ-Glutamyl Carboxylase (GGCX)

Vitamin K obtained from the diet is considered to reach the target tissues via lipid absorption and the transport system [17,18]. Once transferred to the cells of the target tissue, vitamin K is metabolized by redox cycling in the intracellular endoplasmic reticulum body, in a process known as the “vitamin K cycle” [19,20,21]. This series of oxidation-reduction reactions begins with conversion of vitamin K from a stable oxidized form (quinone form) to a hydroquinone form by vitamin K epoxide reductase (VKOR). GGCX carboxylates the glutamic acid residues of vitamin K-dependent proteins (VKDP) to Gla using reduced vitamin K, while simultaneously oxidizing the reduced form of vitamin K to an epoxide form. These reactions that GGCX catalyzes proceed on the GGCX protein molecule using CO_2_ and O_2_; however, the detailed molecular mechanisms are not clear. The epoxide form of vitamin K is reduced by epoxide reductase (vitamin K epoxide reductase complex 1; VKORC1 or vitamin K epoxide reductase complex 1-like 1; VKORC1L1) to a reduced form and then to the reduced hydroquinone form (Figure 2) [22,23,24,25,26]. This reuse system allows for a very small amount of vitamin K in cells to act efficiently as a cofactor of GGCX in the post-translational carboxylation of VKDPs. Warfarin, an oral anticoagulant drug, inhibits VKOR, stops the vitamin K cycle, and prevents the γ-glutamyl-carboxylated (Gla) conversion of the blood coagulation factors, thus inhibiting coagulation (Figure 2). Activity of both GGCX and VKOR are regulated by calumenin [27,28]. Recent evidence has revealed that GGCX is the only enzyme involved in Gla formation, based on structure and function analyses of GGCX at the gene level and animal studies showing that GGCX gene deficiency causes embryonic lethality from systemic bleeding.

Thus, vitamin K acts as a cofactor for GGCX via the vitamin K cycle and exerts physiological effects through its regulation of VKDPs [29]. More than 20 VKDPs have been found. Osteocalcin promotes bone formation, and blood coagulation factors II, VII, IX, and X activate blood coagulation. Matrix Gla protein suppresses cardiovascular calcification, and brain-expressed Gas 6 promotes neural differentiation [29]. GGCX is an enzyme that converts glutamic acid (Glu) residues to Gla residues, so that the Gla-containing proteins can exert various physiological actions such as blood coagulation and bone formation.

Few studies, however, have addressed derivatives targeting GGCX. Vermeer et al. [30] found increased GGCX activity with modifying the side-chain structure of vitamin K to a saturated alkyl side chain with an amide bond. Further modification of the side-chain structure is anticipated from synthesis of vitamin K derivatives that yield stronger GGCX activity. Development of such new derivatives may lead to drug discovery that can enhance the physiological effects associated with GGCX and other factors [31].

## 3. Vitamin K Derivatives Targeting Steroid and Xenobiotic Receptor (SXR) to Mediate Transcriptional Activity

Tabb et al. revealed that MK-4 regulates gene expression as a ligand of the nuclear receptor SXR [32]. SXR (NR1I2) is mainly expressed in the liver and intestine and regulates expression of genes encoding enzymes involved in steroid metabolism and detoxification of xenobiotics and of various drugs [32]. SXR is a human homolog of the mouse pregnane X receptor (PXR), with 95% sequence homology in the N-terminal DNA-binding domain and 73% homology in the C-terminal ligand-binding domain, a characteristic structure of the nuclear receptor [32,33]. When bound to a ligand, SXR forms a heterodimer with retinoid X receptor, and the resulting complex then binds to an SXR responsive element on the target gene promoter via the DNA-binding domain to exert transcription regulation (Figure 3) [34]. In addition to bile acids (e.g., lithocholic acid), drugs such as rifampicin, SR12813, and hyperforin are ligands involved in this process. Among the vitamin K homologs, MK-4 can activate transcription of the SXR target gene *CYP3A4*, as a ligand of SXR [34]. MK-4 plays an important role in osteoblast formation by inducing expression of genes such as matrilin-2 and tsukushi, which are involved in collagen accumulation via SXR [33]. An in vitro study further showed that overexpression of SXR and its activation by MK-4 inhibits proliferation and migration of liver cancer cells [35]. More recently, the data showed that VK_2_ has a differentiation-promoting effect on myeloid progenitors and an anti-apoptotic effect on erythroid progenitors [36].

Thus, MK-4 works via SXR to regulate the expression of various genes at the transcriptional level, resulting in broad physiological effects such as bone formation and liver cancer suppression as well as drug metabolism. However, to date, MK-4 is the only vitamin K homolog known to exert its activities via SXR, and further research is needed to clarify whether other vitamin K homologs act as SXR ligands. X-ray crystal structure analysis of complexes of PXR and ligands (such as rifampicin) have demonstrated that the ligand-binding region of SXR is large, with substantial flexibility [37,38], suggesting that other vitamin K congeners likely could act as SXR ligands. Because the structure of MK-4 can be roughly divided into a 2-methyl-1,4-naphthoquinone ring and a geranylgeranyl side chain, we focused on the features of the ring structure and the side-chain structure of vitamin K and its various homologs. In particular, we investigated the ligand recognition properties of SXR using MK-4 derivatives.

For this purpose, we synthesized various kinds of vitamin K derivatives and compared their biological activities to those of natural homologs. First, we focused on modification of the double bond or methyl group in the side chain of MK-4. Compounds **4**–**13** were synthesized by step-wise saturation of the number of double bonds in the side-chain moiety or deletion of one of the methyl groups (Figure 4A). To investigate the effects of these changes on transcriptional activity, in addition to modification of the side chains, we introduced a deuterium label to monitor the effects [39]. To evaluate the transcriptional activity of each SXR-mediated compound, we used two methods involving *SXR-GAL4* and *CYP3A4* promoters, respectively. Overall, we found significantly decreased transcriptional activity with a decreased number of double bonds compared to MK-4 [40]. This tendency was most apparent in the assay for the *CYP3A4* promoter. Similarly, the transcriptional activity was reduced for compounds with a methyl group deleted.

Based on these results, we predicted that the side chain of MKs would be the key determinant of their biological activity. Therefore, we synthesized new vitamin K derivatives, compounds **14**–**18**, in which isoprene side chains were symmetrically introduced into the naphthoquinone skeleton of vitamin K homologs (Figure 4B) [40], and examined their SXR-mediated transcriptional activity mediated. Although this activity was lower than natural vitamin K homologues, the result showed the greatest transcriptional regulatory activity with compound **16**, in which two geranyl side-chain moieties were introduced to 1,4-naphthoquinine, whereas the activities of **14**, **17**, and **18** gradually decreased. These analyses revealed that the length of the side-chain moiety and the bulkiness significantly affect transcriptional activity of vitamin K derivatives.

We further examined the effect on vitamin K transcriptional activity of differences in the polarity of hydrophilic or lipophilic functional groups introduced to the ω-position of the side chain. Specifically, compounds **19**–**21** were synthesized with introduction of a hydroxy group as a hydrophilic functional group, and compounds **22** and **23** were synthesized with a phenyl group as a hydrophobic functional group (Figure 4C). Compounds **19**–**21** with a hydroxy group showed decreased transcriptional activity and lower than natural vitamin K homologues, whereas the activities of **22** and **23** with the phenyl group significantly increased and were higher than natural vitamin K homologues. In particular, compound **22** with a phenyl group at the ω-side chain of MK-3 showed almost equal activity to that of rifampicin, a known SXR ligand. We also performed a docking simulation between the ligand-binding domain of SXR and the vitamin K derivative, using a Molecular Operating Environment [41]. The result showed that compound **22** fit into the binding pocket of SXR, thanks to its appropriate side-chain length and bulkiness, and that hydrogen bonds formed between the oxygen atom of **22** in the quinone moiety and the His 407 or Ser 247 residue of SXR [42]. MK-3 and MK-4, which showed higher activity, were also expected to exhibit a similar binding state. Thus, these studies clarified that the length and bulkiness of the isoprene side chain are the main determinants of the transcriptional regulatory activity of vitamin K derivates via SXR [39,40,41,42].

## 4. Differentiation-Inducing Action of Neural Stem Cells by Vitamin K and Derivatives

Because MK-4 is present in the brain at a relatively high concentration, it is thought to have important roles in the brain [10]. Vitamin K protects neural cells from oxidative stress; however, a crucial role for vitamin K in the brain has not been elucidated. Neural stem cells engage in continuous self-replication while maintaining the ability to differentiate into neurons and glial cells in the early embryonic and late fetal stages. They can differentiate into neuronal precursor cells and glial precursor cells, and each progenitor cell differentiates into neurons, astrocytes, and oligodendrocytes. The neural stem cells that do not differentiate into neurons differentiate into glial cells before and after birth, at which point differentiation into neurons is complete (Figure 5) [43].

Neuronal progenitor cells differentiate into neurons, while glial progenitor cells differentiate into astrocytes and oligodendrocytes [44]. We recently reported that MKs show weak activity in driving the differentiation of progenitor cells to neuronal cells [45], which depended on the number of isoprene units of the side chain in the vitamin K homolog. If this differentiation activity can be increased by modification of the chemical structure, it would be possible to regulate it with a specific neuronal differentiation inducer based on a low-molecular-weight compound. Such a compound could provide an alternate strategy to conventional gene induction methods used in induced pluripotent stem cells and other types of stem and progenitor cells. Therefore, we focused on the role of vitamin K in the regeneration of neurons, and synthesized vitamin K analogs that could differentiate progenitor cells into neuronal cells.

Because the brain readily incorporates fats, we designed and synthesized a novel vitamin K derivative in which a hydrophobic functional group such as benzene or naphthalene was introduced (Figure 6A). For the synthesis of the target compound, the side-chain moiety was separately synthesized in the same manner as described for the above derivatives and then coupled with 2-methyl-1,4-naphthoquinone. Next, we investigated the differentiation-inducing effect of the obtained compounds in neurons using neural stem cells derived from the cerebrum of 14-day-old mouse embryos. After addition of various compounds to the neural stem cells at 1 μM and 4 days of culture, we applied fluorescence immunostaining and confocal laser microscopy to detect expression of microtubule associated protein 2 (*Map2*) and glial fibrillary acidic protein (*GFAP*), which are differentially expressed in neurons and astrocytes, respectively. In addition, to assess for differences in the differentiation-inducing abilities among the synthesized compounds, we quantified mRNA expression levels in the cells by real-time polymerase chain reaction. Fluorescent immunostaining detected cells that differentiated into neurons, whereas real-time polymerase chain reaction showed that most of the synthesized compounds differed significantly in their differentiation-inducing ability in neurons compared with the control group. In particular, among all compounds tested, the derivative **25b**, in which an *m*-methylphenyl group was introduced to the ω-side chain of MK-3, exhibited the highest activity. In addition, comparison of the relative mRNA expression levels of *Map2* and glial fibrillary acidic protein among the derivatives showed that compound **25b** had selective differentiation-inducing ability in neurons, approximately twice that of the control group. Furthermore, we designed and synthesized compounds **29ab**–**34ab**, in which a substituent such as a fluorine atom or a methyl group was introduced to incorporate a heteroatom at the ω-side chain that could interact with proteins related to neuronal differentiation (Figure 6A). As expected, the compounds containing fluorine atoms exhibited higher Map2 mRNA expression.

To further investigate the specific factors contributing to the differentiation-inducing activity of compound **25b**, we synthesized compounds **35**–**46**, in which *t*-butyl and methyl groups were introduced to the phenyl group at the omega-position of the side chain. Compounds **40** and **46** exhibited the highest activity among the compound group, equal to MK-4. Of interest, compounds **35** and **37**, with the phenyl group introduced at the 2,3- and 3,4-positions of the phenyl group at the end of the MK-2 side chain, clearly suppressed the differentiation of progenitor cells into neurons (Figure 6B) [45,46]. These findings clarified that it is possible to enhance the differentiation-inducing activity of vitamin K by introducing a hydrophobic functional group at the ω-side chain. Moreover, most natural products reported to induce neuronal differentiation, including neuropathiazol, epolactaene, and retinol (retinoic acid) [47,48,49], possess double bonds and phenyl groups in their side-chain moiety, and show certain structural similarities with our synthesized compounds. Based on these findings, we synthesized compounds with other hydrophobic substituents introduced at the end of the side chain and with heteroatoms introduced into the molecule, in anticipation of enabling an interaction with the protein driving neuronal differentiation activity. Detailed elucidation of the precise mechanism of the differentiation-inducing activity of vitamin K derivatives and their other biological actions will help guide the design of compounds with strong activity on proteins of interest.

## 5. Conclusions

Our recent work reviewed here clarifies that vitamin K has a great influence on biological processes, with activities that depend on the different properties of the alkyl side-chain moiety. With further focus on these activities, we expect to reveal the detailed influence of vitamin K on proteins related to neuronal differentiation and gene transcription. In addition to vitamin K, vitamins A, D, and E are all fat-soluble vitamins and have similar side chain structures. These vitamins show a similar biological activity including regulation of transcriptional activity through nuclear receptors. Such similarities are very interesting and will be important information for synthesizing new derivatives of vitamin K in the future. These differences in the alkyl side chain structure ensure that an optimal side chain structure is always present to enhance the target biological activity even though each action is unique to each type of vitamin. At present, we are planning to further investigate the structural specificity of the side chain and to evaluate the structure–activity relationship of the naphthoquinone ring, which has not yet been considered in our research. Overall, the ultimate goal of this line of work is to identify new biologically active compounds based on the structure of vitamin K that could be developed into useful drug compounds.

## Figures and Tables

**Figure 1 ijms-20-03006-f001:**
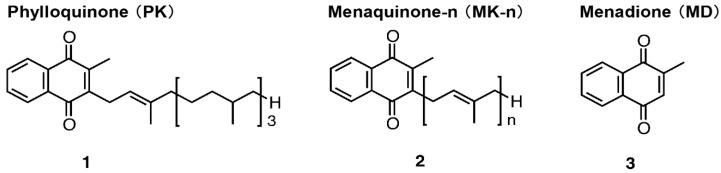
Chemical structure of vitamin K.

**Figure 2 ijms-20-03006-f002:**
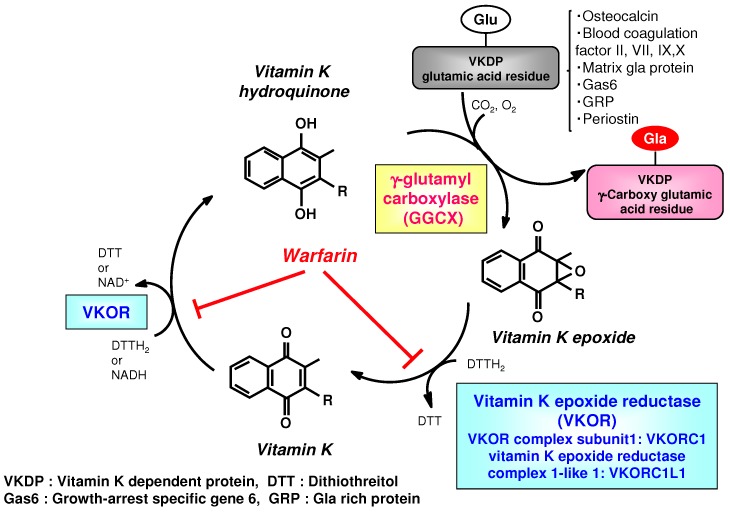
Overview of Vitamin K cycle.

**Figure 3 ijms-20-03006-f003:**
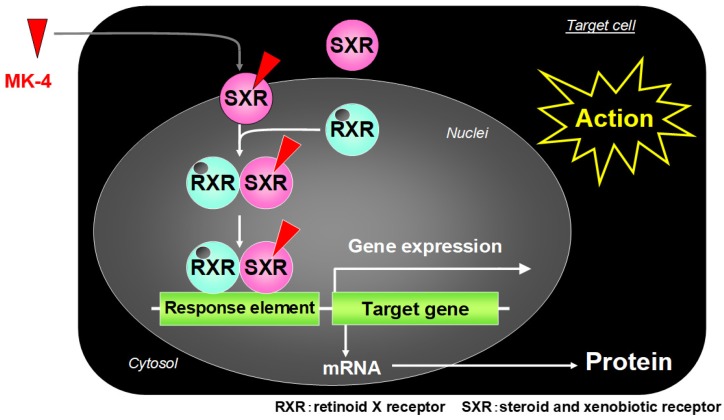
Transcription regulatory action of menaquinone-4 (MK-4) via the nuclear steroid and xenobiotic receptor (SXR).

**Figure 4 ijms-20-03006-f004:**
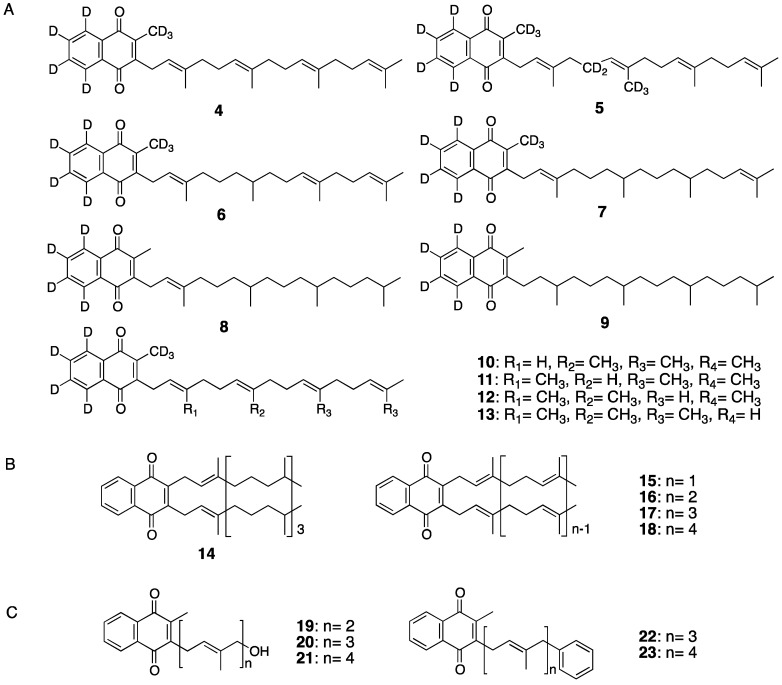
Vitamin K derivatives investigated for SXR-mediated transcriptional regulatory activity. (**A**) Deuterium labeled derivatives, (**B**) Double side chain derivatives, (**C**) Derivatives introduced a hydrophilic or a hydrophobic functional group.

**Figure 5 ijms-20-03006-f005:**
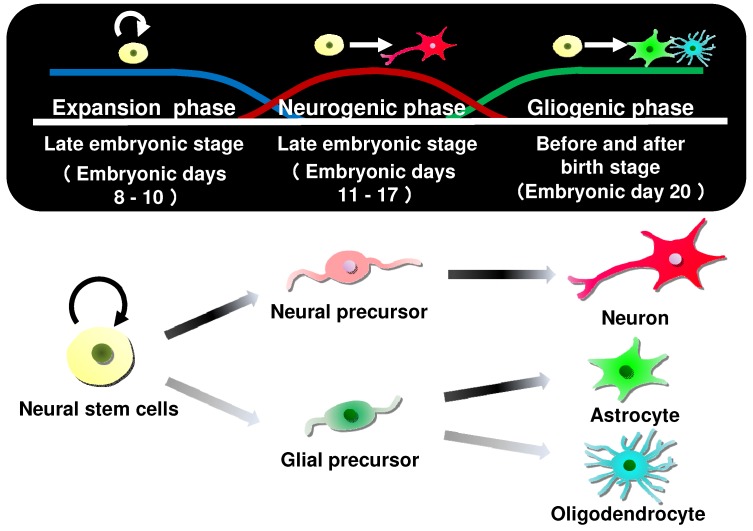
Differentiation of neural stem cells in the brain.

**Figure 6 ijms-20-03006-f006:**
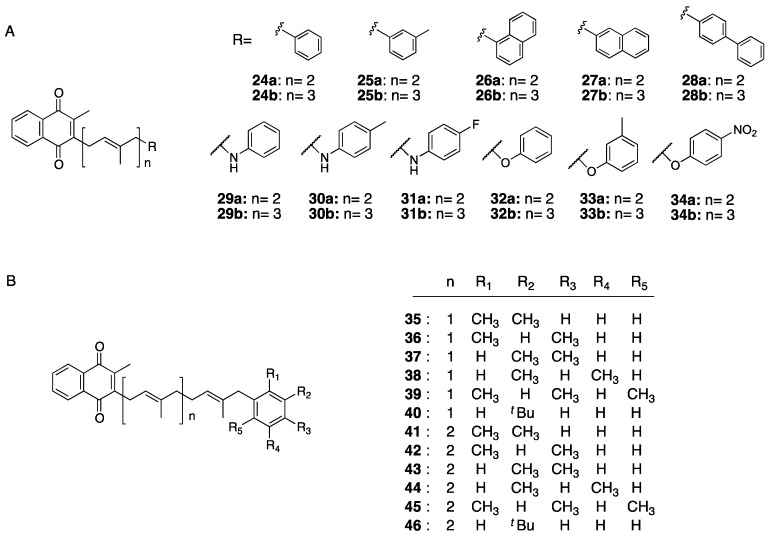
Vitamin K derivatives with neuronal differentiation-inducing activity. (**A**) Derivatives introduced an aromatic ring, or an aromatic ring and a heteroatom, (**B**) Derivatives having a phenyl group with a t-butyl and methyl groups.

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
