# Peer review of "New Aspects of Vitamin K Research with Synthetic Ligands: Transcriptional Activity via SXR and Neural Differentiation Activity"

_ijms, 2019, doi:10.3390/ijms20123006_

Round 1
Reviewer 1 Report
The paper by Suhara Y et al: New roles of vitamin K in transcriptional activity and
neural differentiation revealed by research with synthetic ligands.
The Manuscript is interesting but It needs minor revision:
· Page 1, line 40: The Authors should add others foods where can be found MKn (Booth S 2000)
· Page 2, line46: The Authors are strongly advise better explain respect to MK4 concentrations in human: especially they should discriminate between Japanese population and west people. Furthermore they should specify about great limitation due to lack of a standard method to measure vitamin k. For that, they strongly advise to cite Fusaro et al Vitamin K plasma levels determination in human health. Clin Chem Lab Med. 2017 May 1;55(6):789-799.
· Page 3, Figure 2: should be improved, eg: the authors should add others relevant VKDPs such as Gas6, GRP, Periostin
· Page 3, line 91:The Authors should add others relevant effects of VKDPs on vascular system
· Page 4, line 125: The Authors should write reference relate to this sentence: However, to date, MK-4 is the only vitamin K homolog demonstrated to exertits activities via SXR, and thus further research is needed to clarify whether other vitamin K homologs act as ligands for SXR.
In literature has been highlighted this function to longer MKn than MK4 too (eg MK7 and so on)
· Page 5, line 150-153: The Authors should better explain their result and comparing it to findings Vermeer et al 2017 (see 26 reference)
· Page 6: The Authors should explain if the differentiation-inducing action neural stem cells by vitamin K and derivates can be applied to others type of cells and How
· Page 8: Conclusions should be Recap and especially improved english form
Author Response
Reviewer 1
The paper by Suhara Y et al: New roles of vitamin K in transcriptional activity and neural differentiation revealed by research with synthetic ligands. The Manuscript is interesting but It needs minor revision:
Comment: Page 1, line 40: The Authors should add others foods where can be found MKn (Booth S 2000)
Response: Thank you for your comment. Our corrections are noted below, highlighted in yellow:
Original: Natural vitamin K includes phylloquinone (PK) derived from plants and menaquinones (MK-n) derived from intestinal bacteria and fermented foods [3]
Edited: →Natural vitamin K includes phylloquinone (PK) derived from plants (e.g., green vegetables) and menaquinones (MK-n) derived from intestinal bacteria and fermented foods (e.g., cheeses and the Japanese soybean product natto) [3, 4]
Added reference: 4. Booth SL, Tucker KL, Chen H, Hannan MT, Gagnon DR, Cupples LA, Wilson PW, Ordovas J, Schaefer EJ, Dawson-Hughes B, Kiel DP. Dietary vitamin K intakes are associated with hip fracture but not with bone mineral density in elderly men and women. Am. J. Clin. Nutr., 2000, 71, 1201-1208.
Comment: Page 2, line46: The Authors are strongly advise better explain respect to MK4 concentrations in human: especially they should discriminate between Japanese population and west people. Furthermore they should specify about great limitation due to lack of a standard method to measure vitamin k. For that, they strongly advise to cite Fusaro et al Vitamin K plasma levels determination in human health. Clin Chem Lab Med. 2017 May 1;55(6):789-799.
Response: Thank you for your comment. We have added the following to the text:
The dominant dietary form of vitamin K in the United States and Europe is phylloquinone (90% of daily intake), whereas the major form in Japan is menaquinones (10%), especially menaquinone 7 (MK-7) [5, 6, 7].
Added references: 5. Fusaro M, Gallieni M, Rizzo MA, Stucchi A, Delanaye P, Cavalier E, Moysés RMA, Jorgetti V, Iervasi G, Giannini S, Fabris F, Aghi A, Sella S, Galli F, Viola V, Plebani M. Vitamin K plasma levels determination in human health. Clin. Chem. Lab. Med., 2017, 55, 789-799.
6. Kaneki M, Hodges SJ, Hosoi T, Fujiwara S, Lyons A, Crean SJ, Ishida N, Nakagawa M, Takechi M, Sano Y, Mizuno Y, Hoshino S, Miyao M, Inoue S, Horiki K, Shiraki M, Ouchi Y, Orimo H. Japanese fermented soybean food as the major determinant of the large geographic difference in circulating levels of vitamin K2: possible implications for hip-fracture risk. Nutrition, 2001, 17, 315–21.
7. Booth SL, Suttie JW. Dietary intake and adequacy of vitamin K. J. Nutr., 1998, 128, 785–788.
Comment: Page 3, Figure 2: should be improved, eg: the authors should add others relevant VKDPs such as Gas6, GRP, Periostin
Response: Thank you for your comment. We have adjusted the figure, which now is as presented below:
Fig. 1 Overview of Vitamin K cycle
Comment: Page 3, line 91:The Authors should add others relevant effects of VKDPs on vascular system
Response: Thank you for your comment. We have made the changes highlighted in yellow below:
More than 20 VKDPs have been found. Osteocalcin promotes bone formation, and blood coagulation factors II, VII, IX, and X and proteins C and S activate blood coagulation. Matrix Gla protein suppresses cardiovascular calcification, and brain-expressed Gas 6 promotes neural differentiation [29].
Comment: Page 4, line 125: The Authors should write reference relate to this sentence: However, to date, MK-4 is the only vitamin K homolog demonstrated to exertits activities via SXR, and thus further research is needed to clarify whether other vitamin K homologs act as ligands for SXR.
Response: We have deleted this sentence to avoid misunderstanding for readers.
Comment: In literature has been highlighted this function to longer MKn than MK4 too (eg MK7 and so on)
Response: We have removed this sentence from Figure 3 to avoid misunderstanding for readers because we intended to focus only on MK-4.
Comment: Page 5, line 150-153: The Authors should better explain their result and comparing it to findings Vermeer et al 2017 (see 26 reference)
Response: We have mentioned the findings by Vermeer et al. on page 3, lines 95–97 in section 2. They described about synthetic vitamin K derivatives for GGCX. We think that adding it also to this section would not be a good fit, as this section describes SXR.
Comment: Page 6: The Authors should explain if the differentiation-inducing action neural stem cells by vitamin K and derivates can be applied to others type of cells and How
Response: Thank you for your comment. There has been no report that derivatives induce differentiation of other cells. For this reason, we cannot offer further context or information.
Comment: Page 8: Conclusions should be Recap and especially improved english form
Response: The paper has been professionally edited by a scientific-editing service.

Reviewer 2 Report
The paper by Suhara and Hirota presents a good review of the new physiological actions of vitamin K unveiled using synthetic compounds which have shown to be more potent than the natural vitamin.
The authors have summarized in a concise manner the available data in this area despite a strong emphasis in their own work.
My main concern is in the localization of the references listed in the text, which are often missplaced. For example ref 27 (line 103) should be replaced by ref 28 and those following in that paragraph should be carefully checked to confirm they are not wrongly referred at those locations. Also, in line 144, reference 1 is not adequate to the text where it is inserted. Idem for line 188, ref 43 does not appear to correspond to the text. Maybe ref 44?
Additional comments:
Fig 2: legend should be more explicit. Not clear why the word "cloning" is in the figure... what is meant by that? Also, what about VKD proteins expressed in brain? it should be relevant for this paper to mention them. Also maybe in the discussion?
Sentence in lines 89 to 91: not clear
Author Response
Reviewer 2
The paper by Suhara and Hirota presents a good review of the new physiological actions of vitamin K unveiled using synthetic compounds which have shown to be more potent than the natural vitamin.
The authors have summarized in a concise manner the available data in this area despite a strong emphasis in their own work.
Comment: My main concern is in the localization of the references listed in the text, which are often missplaced. For example ref 27 (line 103) should be replaced by ref 28 and those following in that paragraph should be carefully checked to confirm they are not wrongly referred at those locations. Also, in line 144, reference 1 is not adequate to the text where it is inserted. Idem for line 188, ref 43 does not appear to correspond to the text. Maybe ref 44?
Response: We have corrected the locations of the reference numbers. Thank you very much for your suggestions.
Additional comments:
Comment: Fig 2: legend should be more explicit. Not clear why the word "cloning" is in the figure... what is meant by that? Also, what about VKD proteins expressed in brain? it should be relevant for this paper to mention them. Also maybe in the discussion?
Sentence in lines 89 to 91: not clear
Response: Thank you for your comment. The notation of cloning is our mistake, which we have corrected as shown in Fig. 1 above. In addition, VKDP in the brain has been known as Gas6 so far. We also have added the sentence highlighted below:
The major VKDPs include osteocalcin and matrix Gla protein, which are distributed in blood coagulation factors II, VII, IX, X, protein C and S, and the bone.
→ More than 20 VKDPs have been found. Osteocalcin promotes bone formation, and blood coagulation factors II, VII, IX, and X and proteins C and S activate blood coagulation. Matrix Gla protein suppresses cardiovascular calcification, and brain-expressed Gas 6 promotes neural differentiation [29].
Reviewer 3 Report
This manuscript addresses the involvement of vitamin k in different biological processes through studies using vitamin K derivatives, an issue of great interest with potential clinical implications.
However, the paper needs vast improvement and a careful revision for clarity, accuracy and linearity to represent a valuable review of data in literature.
General comments:
From the abstract to the conclusions, it is not clear if the authors aimed at presenting a literature review or a summary of their studies on the field. If it is the former, this should be clearly stated in the aims of the review. Coherently, authors should devote more attention at the studies reported by the international literature and not detail only the research previously conducted by themselves in the field. In this regard, I will also suggest to use an impersonal form and not “we observed” or “we use” etc.
The detailed chemistry of the single vitamin K derivatives studied by authors is provided, but the experimental model and the observed or measured biological effects need a more clear, linear and synthetic presentation. Only major results should be presented and discussed.
I have the following major and minor comments for the authors to address.
Title
The title does not appear conceptually adequate: “transcriptional activity” and the “neural differentiation” are indicated as distinct new roles of vitamin K.
The regulation of the transcription of specific genes could represents a mechanism through which vitamin K might be involved or plays a role in different biological processes (i.e. osteoblast formation, proliferation and migration of liver cancer cells) including also the neuronal differentiation. Indeed, the involvement of vitamin K in the regulation of the transcription of specific genes, coding for proteins regulating the differentiation process cannot be excluded.
This concern should be discussed by authors also in the text.
Abstract
Line 28: modify as follows “agonist of steroid and xenobiotic nuclear receptor”
Introduction
Line 37: insert bleeding in “treatment and prevention of bleeding and osteoporosis.”
Lines 52-62 This part of the introduction should be modified, taking in consideration my first concern “literature review or author’s studies”.
Section 2
The overall description of the vitamin K cycle needs more clarity. The involved enzymatic activities should be correctly reported.
The functional/ significance of the gamma- carboxylation must be mentioned, that consists in the calcium binding by the Gla- residues essential for the proteins (VKDP) to fulfill their physiological functions.
Line 70: GGCX catalyzes the post-translational carboxylation of glutamic acid residues, it is not an oxidation of glutamic acid residues.
Line 73: Reaction solution ? It is unclear.
Line 74: the epoxide form is reduced by epoxide reductase, not “is converted to a oxidized form by epoxide reductase”.
Line 76: insert hydroquinone “and then to the reduced hydroquinone form.”
Line 78: “to efficiently activate” substitute with “to efficiently act as cofactor of GGCX in the post-translational carboxylation of VKDPs.
Line 80: factors
Lines 88-93: uncorrected sentence structure, a rewriting is required.
“these proteins are activated by GGCX”: it is not an activation, see comments above.
Lines 94-99: “Derivatives targeting GGCX….. ” Given that the paper is a review, a more expanded report of studies on derivatives targeting GGCX and their effects should be provided. This is also expected based on the title of the section 2.
Figure 2. The title should be changed: the vitamin K cycle as a whole contains the redox reactions, is not “responsible for the redox reaction”.
An explicative legend containing the expansion of the abbreviations is also suggested for clarity. In the box on the left it is reported VKQR, is it VKOR? DTTH2? does it refer to a reducing equivalent donor (but not “in vivo”) ?
Section 3
Line 102: “In addition to the Gla-mediated action of VKDP in the vitamin K cycle”. In this form the sentence means that vitamin K dependent proteins (VKDP) have a role in the vitamin K cycle. Please rewrite the sentence.
Line 103 : in the text reference number 27 is the paper of Tabb, while in the list of references Tabb is the number 28. The reference 27 (Stafford) regards the vitamin K cycle and thus should be reported in the upper 102 line.
Lines 104-105: “SXR ….regulated the expression of xenobiotics ….” This is wrong. SXR regulates the expression of genes coding for enzymes involved in the metabolism of steroids and in detoxification of xenobiotics and of various drugs.
Lines 114: CYP3A4, not CXP3A4. Please report also the function of this gene.
Buitenhuis et al (ref 31) investigated the vitamin K1, K2 and K3 as cofactor of GGCX and not the transcription regulation activity of MK-4. Buitenhuis et al could be cited in the text where the targeting of GGCX is reported.
Line115: matrilin-2, not matllirin2
Line118: “ MK4 was reported to inhibit the differentiation…” is uncorrected. In fact, as reported at the end of the introduction of the paper of Sada et al (ref33): “the data showed that VK2 has a differentiation-promoting effect on myeloid progenitors and an anti-apoptotic effect on erythroid progenitors”.
Lines 118-120 Which genes were suggested to be the target of the regulatory action of MK-4 on the cells? In other words, for which genes the mRNA expression levels were increased after MK-4 treatment? Or proteins?
Figure 3: the sentence, “PK and MK did not bind” (not “have not bound”), should be inserted in the text (line 125-126)
Lines 139-140. “The conversion of vitamin K derivatives into MK-4 was investigated”. This is unclear in a context of transcriptional regulation.
Lines 142-143: a brief description of the two methods/models/assays should be provided to help the readers understand. Were these two models used in parallel to test all the derivatives reported in figure 4 ? Moreover, is the sentence “This tendency was most strongly apparent in the assay for the CYP3A4 promoter” true for all the derivatives reported in figure 4 (A,B,C)?
Fig 4- This figure does not report the activity thus the title should be modified “Vitamin K derivatives investigated for SXR-mediated transcriptional regulatory activity”.
Line 153: “transcriptional regulatory activity”
Line 168: “MK-3 and MK-4, which showed higher activity……..” if compared with which derivatives? Overall it is not clear if some of the several derivatives reported by authors exhibited a stronger regulatory activity than the natural vitamin K.
Line 170: “ ….determinants for the transcriptional……” . The adjective “regulatory” has to be added “transcriptional regulatory activity”
Line 181: “early embryonic stage”, not brains
Line 203: in addition to the symbols, the name of the proteins should be reported for clarity
Line 213: “into neurons” not “on neurons”
Line 214: which was the “control group” ?
Lines 217-218: “…the neuronal differentiation activity……” This is not a very informative sentence. In terms of mRNA expression levels was it higher, lower or similar to that of the other derivatives?
Lines 229-231: this sentence is quite vague for a review and unclear. Do authors refer to a specific study (paper)? Or is this sentence, together with the subsequent, part of the conclusions?
Section 5
The mention of vitamins A, D and E is interesting. It is known that vitamins A, D and E present a regulatory transcriptional activity when bound to nuclear receptors, thus authors should stress and better discuss this issue in relation to vitamin K.
The prospects of the research seem to refer only to future investigations by the authors. Please see the general comment.
References Please carefully check references and their exact position in the main text and the coherent numbering.
Author Response
Reviewer 3
This manuscript addresses the involvement of vitamin k in different biological processes through studies using vitamin K derivatives, an issue of great interest with potential clinical implications.
However, the paper needs vast improvement and a careful revision for clarity, accuracy and linearity to represent a valuable review of data in literature.
General comments:
Comment: From the abstract to the conclusions, it is not clear if the authors aimed at presenting a literature review or a summary of their studies on the field. If it is the former, this should be clearly stated in the aims of the review. Coherently, authors should devote more attention at the studies reported by the international literature and not detail only the research previously conducted by themselves in the field. In this regard, I will also suggest to use an impersonal form and not “we observed” or “we use” etc.
Response: This review mainly offers a summary of our study. Because derivative research in vitamin K is scarce, we developed this review paper. The journal appears to allow authors to use first person, which facilitates sentences in the active voice.
Comment: The detailed chemistry of the single vitamin K derivatives studied by authors is provided, but the experimental model and the observed or measured biological effects need a more clear, linear and synthetic presentation. Only major results should be presented and discussed.
Response: We have edited part of this sentence.
I have the following major and minor comments for the authors to address.
Title
The title does not appear conceptually adequate: “transcriptional activity” and the “neural differentiation” are indicated as distinct new roles of vitamin K.
The regulation of the transcription of specific genes could represents a mechanism through which vitamin K might be involved or plays a role in different biological processes (i.e. osteoblast formation, proliferation and migration of liver cancer cells) including also the neuronal differentiation. Indeed, the involvement of vitamin K in the regulation of the transcription of specific genes, coding for proteins regulating the differentiation process cannot be excluded.
Comment: This concern should be discussed by authors also in the text.
Response: Thank you for your comment. We have added the following yellow-highlighted heading:
New aspects of vitamin K research with synthetic ligands: transcriptional activity via SXR and neural differentiation activity
Abstract
Comment: Line 28: modify as follows “agonist of steroid and xenobiotic nuclear receptor”
Response: We have made the changes that the reviewer suggested.
Introduction
Comment: Line 37: insert bleeding in “treatment and prevention of bleeding and osteoporosis.”
Response: We have made this change. Thank you very much for your suggestion.
Comment: Lines 52-62 This part of the introduction should be modified, taking in consideration my first concern “literature review or author’s studies”.
Response: We have clarified that this manuscript as a summary of our work.
Section 2
Comment: The overall description of the vitamin K cycle needs more clarity. The involved enzymatic activities should be correctly reported.
The functional/ significance of the gamma- carboxylation must be mentioned, that consists in the calcium binding by the Gla- residues essential for the proteins (VKDP) to fulfill their physiological functions.
Response: The function/significance of gamma-carboxylation was fully described in the context of the importance of VKDP.
Comment: Line 70: GGCX catalyzes the post-translational carboxylation of glutamic acid residues, it is not an oxidation of glutamic acid residues.
Response: As noted, it should be “carboxylation,” and we have made this correction.
Comment: Line 73: Reaction solution ? It is unclear.
Response: This was an error, and we have deleted it.
Comment: Line 74: the epoxide form is reduced by epoxide reductase, not “is converted to a oxidized form by epoxide reductase”.
Line 76: insert hydroquinone “and then to the reduced hydroquinone form.”
Line 78: “to efficiently activate” substitute with “to efficiently act as cofactor of GGCX in the post-translational carboxylation of VKDPs.
Line 80: factors
Response: We have corrected this phrasing and appreciate your comments.
Comment: Lines 88-93: uncorrected sentence structure, a rewriting is required.
“these proteins are activated by GGCX”: it is not an activation, see comments above.
Response: γ-Carboxylation is considered to be “activation” because it has the effect of converting proteins from an inactive to an active form.
Comment: Lines 94-99: “Derivatives targeting GGCX….. ” Given that the paper is a review, a more expanded report of studies on derivatives targeting GGCX and their effects should be provided. This is also expected based on the title of the section 2.
Response: As described in the manuscript, few compounds have been reported to increase GGCX activity. Therefore, there is not much room to discuss derivatives for GGCX and their effects, which we feel is beyond the scope of this review.
Comment: Figure 2. The title should be changed: the vitamin K cycle as a whole contains the redox reactions, is not “responsible for the redox reaction”.
An explicative legend containing the expansion of the abbreviations is also suggested for clarity. In the box on the left it is reported VKQR, is it VKOR? DTTH2? does it refer to a reducing equivalent donor (but not “in vivo”) ?
Response: Thank you for your comment. The notation of cloning is our mistake, which we have corrected, as show in Figure 1 above.
Section 3
Comment: Line 102: “In addition to the Gla-mediated action of VKDP in the vitamin K cycle”. In this form the sentence means that vitamin K dependent proteins (VKDP) have a role in the vitamin K cycle. Please rewrite the sentence.
Response: We have deleted this sentence, as it is not necessary in this section.
Comment: Line 103 : in the text reference number 27 is the paper of Tabb, while in the list of references Tabb is the number 28. The reference 27 (Stafford) regards the vitamin K cycle and thus should be reported in the upper 102 line.
Response: We have corrected the reference numbers.
Comment: Lines 104-105: “SXR ….regulated the expression of xenobiotics ….” This is wrong. SXR regulates the expression of genes coding for enzymes involved in the metabolism of steroids and in detoxification of xenobiotics and of various drugs.
Response: Thank you for your comment. We have rewritten as follows:
regulates expression of genes encoding enzymes involved in steroid metabolism and detoxification of xenobiotics and of various drugs [32].
Comment: Lines 114: CYP3A4, not CXP3A4. Please report also the function of this gene.
Buitenhuis et al (ref 31) investigated the vitamin K1, K2 and K3 as cofactor of GGCX and not the transcription regulation activity of MK-4. Buitenhuis et al could be cited in the text where the targeting of GGCX is reported.
Response: We have corrected CYP3A4 and the reference number for the publication by Buitenhuis et al.
Comment: Line115: matrilin-2, not matllirin2
Response: We have made this correction. Thank you for your suggestions.
Comment: Line118: “ MK4 was reported to inhibit the differentiation…” is uncorrected. In fact, as reported at the end of the introduction of the paper of Sada et al (ref33): “the data showed that VK2 has a differentiation-promoting effect on myeloid progenitors and an anti-apoptotic effect on erythroid progenitors”.
Response: Thank you for your comment. We have deleted this part because it would be confusing and is not directly related to the content.
Comment: Lines 118-120 Which genes were suggested to be the target of the regulatory action of MK-4 on the cells? In other words, for which genes the mRNA expression levels were increased after MK-4 treatment? Or proteins?
Response: Thank you for your comment. We have deleted this part because it would be confusing and is not directly related to the content.
Comment: Figure 3: the sentence, “PK and MK did not bind” (not “have not bound”), should be inserted in the text (line 125-126)
Response: The sentence has been corrected as suggested.
Comment: Lines 139-140. “The conversion of vitamin K derivatives into MK-4 was investigated”. This is unclear in a context of transcriptional regulation.
Response: The reason we used the phrasing, “The conversion of vitamin K derivatives into MK-4 was investigated,” was that we synthesized deuterated labeled vitamin K derivatives. We described it because the reader may wonder why we synthesized deuterated labels.
Comment: Lines 142-143: a brief description of the two methods/models/assays should be provided to help the readers understand. Were these two models used in parallel to test all the derivatives reported in figure 4 ? Moreover, is the sentence “This tendency was most strongly apparent in the assay for the CYP3A4 promoter” true for all the derivatives reported in figure 4 (A,B,C)?
Response: The findings that a lower number of double bonds in the side chain lower the tendency for transcriptional activity is particularly strong in the assay of CYP3A4 promoter assay and refers only to the compound of Figure 4A, not to B and C.
Comment: Fig 4- This figure does not report the activity thus the title should be modified “Vitamin K derivatives investigated for SXR-mediated transcriptional regulatory activity”.
Response: We corrected the title of the figure as the reviewer suggested. Thank you very much for pointing it out.
Comment: Line 153: “transcriptional regulatory activity”
Response: We have replaced “transcriptional activity” with “transcriptional regulatory activity.”
Comment: Line 168: “MK-3 and MK-4, which showed higher activity……..” if compared with which derivatives? Overall it is not clear if some of the several derivatives reported by authors exhibited a stronger regulatory activity than the natural vitamin K.
Response: We meant “MK-3 and MK-4, which showed higher activity among natural vitamin K homologues,” rather than to compare with derivatives. We have corrected the sentence.
Comment: Line 170: “ ….determinants for the transcriptional……” . The adjective “regulatory” has to be added “transcriptional regulatory activity”
Response: We have added the adjective “regulatory” to the sentence as a reviewer suggested.
Comment: Line 181: “early embryonic stage”, not brains
Response: Thank you for your comment. The notation of cloning is our mistake. It has been corrected.
Comment: Line 203: in addition to the symbols, the name of the proteins should be reported for clarity
Response: Thank you for your comment. The notation of cloning is our mistake and has been corrected.
Comment: Line 213: “into neurons” not “on neurons”
Response: We have corrected it. Thank you very much for your suggestion.
Comment: Line 214: which was the “control group” ?
Response: This “control group” means “ethanol addition group.” When we investigated the activity of the vitamin K derivatives, we usually used ethanol instead of vitamin K compounds for addition to cells.
Comment: Lines 217-218: “…the neuronal differentiation activity……” This is not a very informative sentence. In terms of mRNA expression levels was it higher, lower or similar to that of the other derivatives?
Response: We changed the sentence to “As expected, the compounds containing fluorine atoms exhibited higher Map2 mRNA expression.”
Comment: Lines 229-231: this sentence is quite vague for a review and unclear. Do authors refer to a specific study (paper)? Or is this sentence, together with the subsequent, part of the conclusions?
Response: Because this review is a summary of our study, we refer often to our specific work.
Section 5
Comment: The mention of vitamins A, D and E is interesting. It is known that vitamins A, D and E present a regulatory transcriptional activity when bound to nuclear receptors, thus authors should stress and better discuss this issue in relation to vitamin K.
The prospects of the research seem to refer only to future investigations by the authors. Please see the general comment.
Response: Thank you for your comment. To describe the conclusion simply, we have eliminated the following:
In addition to vitamin K, vitamins A, D, and E also have double bonds in the side chain moiety. These differences in the alkyl side chain structure ensure that an optimal side chain structure is always present to enhance the target biological activity even though each action is unique to each type of vitamin. At present, we are planning to further investigate the structural specificity of the side chain and to evaluate the structure-activity relationship of the naphthoquinone ring, which has not yet been considered in our research. Overall, the ultimate goal of this line of work is to identify new biologically active compounds based on the structure of vitamin K that could be developed into useful drug compounds.
Comment: References Please carefully check references and their exact position in the main text and the coherent numbering.
Response: We have checked all references and their numbers.

Round 2
Reviewer 3 Report
The authors addressed some of my concerns. Some changes have been only indicated in the response to the reviewer, but have not been reported in the text of the manuscript. Some uncorrected sentences still remain.
Some issues have not been discussed by authors:
1) The regulation of the transcription of specific genes could represents the mechanism through which vitamin K might be involved or plays a role in different biological processes, such as osteoblast formation, proliferation and migration of liver cancer cells and neuronal differentiation.
2) In the “Conclusion” of the first submission authors mentioned vitamins A, D and E. This is of interest, in that vitamins A, D and E present a regulatory transcriptional activity when bound to nuclear receptors. The authors would discuss this in relation to vitamin K role, rather than remove this observation from the text.
Abstract: At the end of the abstract the authors claim that “ …..based in research with synthetic ligands, which show stronger physiological activity than natural vitamin K”, but from the text it is not clear which synthetic ligands were “better” than natural vitamin K.
Section 2
The description of the vitamin K cycle and of the involved enzymatic activities still contains errors.
Line 78: the name of the enzyme is “vitamin K epoxide reductase (VKOR), not “vitamin K oxidation reductase”.
Line 85: the epoxide form (KO) is reduced by epoxide reductase to vitamin K, not “is converted to a oxidized form”. This is a two steps reduction: 1) from vitamin K epoxide (KO) to vitamin K, 2) from vitamin K to vitamin K hydroquinone.
Line 90: Please rephrase the sentence concerning calumenin. Activity of both GGCX and VKOR are regulated by calumenin.
Lines 98 and 102: “activation of VKDPs” , “these proteins are activated by GGCX”, “Glu residue are converted to an activated form”.
I would stress again that gamma- carboxylation does not activate proteins. Gamma- carboxylation of Glu residues potentiates the calcium binding properties, which is essential for the proteins (VKDPs) to fulfill their physiological functions in several biological processes, such as blood coagulation and bone formation.
For most of the VKDPs (i.e. coagulation VKDPs) the activation is achieved through limited proteolysis of the zymogen into the active form.
Lines 102-103: Please correct the sentence by using the plural: several glutamic acid residues are converted to Gla residues, the proteins exert various….
Fig 2. The title has been modified, but the other changes in figure 2 suggested by the present reviewer have not been introduced. In the box on the left VKR not VKQR, cloning annotation in the box on the right should be removed.
An explicative legend containing the expansion of the abbreviations (i.e. DTTH2) a reducing equivalent donor in “in vitro” assay is also suggested for clarity.
Section 3
Lines 113: the sentence “In addition to the Gla-mediated action ………” has not been removed as instead stated by authors.
Lines 126-127: For the sentence “MK4 plays an important role in osteoblast formation by……….” the paper of Buitenhuis et al (1990) has been reported as reference. However, as I previously pointed out, the paper of Buitenhuis et al is improperly cited at this point, because it does not refer to MK4 transcriptional activity.
Lines 118-120 of the first submission: “ MK4 was reported to inhibit the differentiation…” . I previously suggested that authors correct the sentence according to the reference Sada et al: “the data showed that VK2 has a differentiation-promoting effect on myeloid progenitors and an anti-apoptotic effect on erythroid progenitors”. The sentence (lines 118-120 of the first submission) should have been corrected, not removed by authors.
Line 161 “the greatest transcriptional activity”, not “the most transcriptional activity”.
Fig 4- The title of figure has not been modified in the manuscript as suggested.
Section 3 (page 5) and Section 4 (pages 7 and 8): It is not clear which of the several reported derivatives exhibited a stronger transcriptional regulatory activity or differentiation induction activity than the natural vitamin K. Authors should clarify this aspect, since at the end of the abstract they claimed that “ …..based in research with synthetic ligands, which show stronger physiological activity than natural vitamin K”.
Line 218: please add the symbol of the gene (GFAP).
It is curious that the authors use the same sentence “The notation of cloning is our mistake, which we have corrected” to answer three different questions.
Conclusion: In the “Conclusion” of the first submission authors mentioned vitamins A, D and E. This is of interest, in that vitamins A, D and E present a regulatory transcriptional activity when bound to nuclear receptors. The authors would discuss this in relation to vitamin K role, rather than remove this observation from the text. This would improve their conclusion.
Author Response
The authors addressed some of my concerns. Some changes have been only indicated in the response to the reviewer, but have not been reported in the text of the manuscript. Some uncorrected sentences still remain.
Some issues have not been discussed by authors:
1) The regulation of the transcription of specific genes could represents the mechanism through which vitamin K might be involved or plays a role in different biological processes, such as osteoblast formation, proliferation and migration of liver cancer cells and neuronal differentiation.
2) In the “Conclusion” of the first submission authors mentioned vitamins A, D and E. This is of interest, in that vitamins A, D and E present a regulatory transcriptional activity when bound to nuclear receptors. The authors would discuss this in relation to vitamin K role, rather than remove this observation from the text.
Abstract:
Comment: At the end of the abstract the authors claim that “ …..based in research with synthetic ligands, which show stronger physiological activity than natural vitamin K”, but from the text it is not clear which synthetic ligands were “better” than natural vitamin K.
Response:
We thought it was not good idea to represent the compounds by their compound number because it cannot be determined from the abstract alone for readers. Moreover, in the short text of the abstract, it is not possible to accurately explain the compounds with strong activity. Therefore, we replaced the sentence as “some of compounds which phenyl groups were introduced to the side chain show better...”.
Section 2
The description of the vitamin K cycle and of the involved enzymatic activities still contains errors.
Comment: Line 78: the name of the enzyme is “vitamin K epoxide reductase (VKOR), not “vitamin K oxidation reductase”.
Response: Thank you for your comment. We changed vitamin K oxidation reductase to vitamin K epoxide reductase.
Line 85: the epoxide form (KO) is reduced by epoxide reductase to vitamin K, not “is converted to a oxidized form”. This is a two steps reduction: 1) from vitamin K epoxide (KO) to vitamin K, 2) from vitamin K to vitamin K hydroquinone.
Response: Thank you for your comment. We changed oxidized form to reduced form.
Line 90: Please rephrase the sentence concerning calumenin. Activity of both GGCX and VKOR are regulated by calumenin.
Response: Thank you for your comment. We changed "The vitamin K cycle regulates the activities of both GGCX and VKOR via calumenin" to " Activity of both GGCX and VKOR are regulated by calumenin ".
Lines 98 and 102: “activation of VKDPs” , “these proteins are activated by GGCX”, “Glu residue are converted to an activated form”.
I would stress again that gamma- carboxylation does not activate proteins. Gamma- carboxylation of Glu residues potentiates the calcium binding properties, which is essential for the proteins (VKDPs) to fulfill their physiological functions in several biological processes, such as blood coagulation and bone formation.
For most of the VKDPs (i.e. coagulation VKDPs) the activation is achieved through limited proteolysis of the zymogen into the active form.
Response: Thank you for your comment.
Lines 98 “activation of VKDPs”→ “VKDPs”
Lines 100 “GGCX activates these proteins, with its glutamic acid (Glu) residue converted to an active form with Gla”→ “GGCX is an enzyme that converts glutamic acid (Glu) residue to Gla residue.
Lines 102-103: Please correct the sentence by using the plural: several glutamic acid residues are converted to Gla residues, the proteins exert various….
Response: Thank you for your comment. We changed "The vitamin K cycle regulates the activities of both GGCX and VKOR via calumenin" to " Activity of both GGCX and VKOR are regulated by calumenin ".
Fig 2. The title has been modified, but the other changes in figure 2 suggested by the present reviewer have not been introduced. In the box on the left VKR not VKQR, cloning annotation in the box on the right should be removed.
An explicative legend containing the expansion of the abbreviations (i.e. DTTH2) a reducing equivalent donor in “in vitro” assay is also suggested for clarity.
Response: Thank you for your comment. We corrected as follows.
Figure 2. Overview of Vitamin K cycle
Section 3
Lines 113: the sentence “In addition to the Gla-mediated action ………” has not been removed as instead stated by authors.
Response:
Thank you for your comment. We missed to remove this sentence.
Lines 126-127: For the sentence “MK4 plays an important role in osteoblast formation by……….” the paper of Buitenhuis et al (1990) has been reported as reference. However, as I previously pointed out, the paper of Buitenhuis et al is improperly cited at this point, because it does not refer to MK4 transcriptional activity.
Response:
We replaced the paper of Buitenhuis at al to Inoue et al. described MK-4 transcriptional activity via SXR.
MK-4 plays an important role in osteoblast formation by inducing expression of genes such as matrilin-2 and tsukushi, which are involved in collagen accumulation via SXR [35].
→MK-4 plays an important role in osteoblast formation by inducing expression of genes such as matrilin-2 and tsukushi, which are involved in collagen accumulation via SXR [33].
Lines 118-120 of the first submission: “ MK4 was reported to inhibit the differentiation…” . I previously suggested that authors correct the sentence according to the reference Sada et al: “the data showed that VK2 has a differentiation-promoting effect on myeloid progenitors and an anti-apoptotic effect on erythroid progenitors”. The sentence (lines 118-120 of the first submission) should have been corrected, not removed by authors.
Response: Thank you for your comment. We added the following sentence.
More recently, the data showed that VK2 has a differentiation-promoting effect on myeloid progenitors and an anti-apoptotic effect on erythroid progenitors [36].
36. Sada E, Abe Y, Ohba R, Tachikawa Y, Nagasawa E, Shiratsuchi M, Takayanagi R. Vitamin K2 modulates differentiation and apoptosis of both myeloid and erythroid lineages. Eur. J. Haematol., 2010, 85, 538-548.
Line 161 “the greatest transcriptional activity”, not “the most transcriptional activity”.
Response: Thank you for your comment. We corrected the sentence.
Fig 4- The title of figure has not been modified in the manuscript as suggested.
Response: We missed to change the title of figures.
Section 3 (page 5) and Section 4 (pages 7 and 8): It is not clear which of the several reported derivatives exhibited a stronger transcriptional regulatory activity or differentiation induction activity than the natural vitamin K. Authors should clarify this aspect, since at the end of the abstract they claimed that “ …..based in research with synthetic ligands, which show stronger physiological activity than natural vitamin K”.
Response:
We added the words to the sentence to clarify which compounds show weaker or stronger activity compared to natural vitamin K homologues as follows,
Page 5,line 167 “compared to MK-4”
Page 5, line 174-175 “Although those activity was lower than natural vitamin K homologues, the...”
Page 5, line 185-187 “ and lower than natural vitamin K homologues” “ and higher than natural vitamin K homologues”
Page 8, line 262 “Compounds 40 and 46 exhibited the highest activity among the compound group and equal to MK-4.”
Line 218: please add the symbol of the gene (GFAP).
Thank you for your comment. We added the following symbol.
glial fibrillary acidic protein (GFAP)
It is curious that the authors use the same sentence “The notation of cloning is our mistake, which we have corrected” to answer three different questions.
Response: We apologize for writing an incorrect reply. The following has been fixed.
Comment: Line 170: “ ….determinants for the transcriptional……” . The adjective “regulatory” has to be added “transcriptional regulatory activity”
Response: We added the adjective “regulatory” to the sentence as a reviewer suggested.
Comment: Line 181: “early embryonic stage”, not brains
Response: We corrected the words. We appreciate your correction.
Comment: Line 203: in addition to the symbols, the name of the proteins should be reported for clarity
Response: We corrected the words. We appreciate your correction.
Conclusion: In the “Conclusion” of the first submission authors mentioned vitamins A, D and E. This is of interest, in that vitamins A, D and E present a regulatory transcriptional activity when bound to nuclear receptors. The authors would discuss this in relation to vitamin K role, rather than remove this observation from the text. This would improve their conclusion.
Response:
We added appropriate comments to the sentence as follow.
Our recent work reviewed here clarifies that vitamin K has a great influence on biological processes, with activities that depend on the different properties of the alkyl side-chain moiety. With further focus on these activities, we expect to reveal the detailed influence of vitamin K on proteins related to neuronal differentiation and gene transcription. In addition to vitamin K, vitamins A, D, and E also have double bonds in the side chain moiety. These differences in the alkyl side chain structure ensure that an optimal side chain structure is always present to enhance the target biological activity even though each action is unique to each type of vitamin. At present, we are planning to further investigate the structural specificity of the side chain and to evaluate the structure-activity relationship of the naphthoquinone ring, which has not yet been considered in our research. Furthermore, in order to elucidate the action mechanism of vitamin K, to clarify the vitamin K binding protein involved in neuronal differentiation-inducing activity is needed. Overall, the ultimate goal of this line of work is to identify new biologically active compounds based on the structure of vitamin K that could be developed into useful drug compounds.

Round 3
Reviewer 3 Report
Some comments were not correctly addressed by authors, thus uncorrected sentences and errors still remain despite two previous revisions. Major wrongness concerns the vitamin K cycle.
Abstract:
Lines 32-33: This part of the final sentence is not properly constructed. Please rephrase.
Section 2
Line 80: The name of the enzyme has not been corrected. The name is vitamin K epoxide reductase, not vitamin K oxidation reductase.
Lines 81-82: Understandably during revision the authors deleted “vitamin K dependent protein” and replaced by vitamin K epoxide reductase. Please correct this completely wrong sentence.
Please use the plural: proteins
Line 87: a reduced form
Lines 105: Factors II, VII, IX and X promote the blood coagulation, but protein C and S are inhibitors of the blood coagulation, thus they do not activate the coagulation. This part of the sentence needs to be coherently corrected.
Lines 107-108: As previously recommended to the authors, use the plural for residues, in that several Glu residues are coverted to Gla.
Line 109: “so that it…..” : is wrong. Please modify as follows “ so that the Gla-containing proteins can exert various ……..”
Section 3
Line 168 “…those activity was lower “, please replace by “this activity was…..”
Line 181: “…..increased and higher than…….”. please add “…….increased and were higher than….”
Line 250”……comound..”, please correct “compound”
Conclusion: Authors reintroduced the sentence including vitamins A, D and E. As I previously suggested, authors should additionally report that vitamins A, D and E, which show some structural similarities with vitamin K, present a regulatory transcriptional activity when bound to nuclear receptors, as similarly observed for vitamin K.
This is an interesting aspect which would improve the Conclusions.
Lines 279-281 The sentence “Furthermore, in order to …………..”, is overall not correctly written, and should be removed as the concept is already reported at lines 271-273.
The same response “We changed "The vitamin K cycle regulates the activities of both GGCX and VKOR via calumenin" to " Activity of both GGCX and VKOR are regulated by calumenin ", has been used to address two different reviewer’ comments.
Author Response
Reviewer 3
Some comments were not correctly addressed by authors, thus uncorrected sentences and errors still remain despite two previous revisions. Major wrongness concerns the vitamin K cycle.
Response: Thank you for your comment. Sorry, there are a lot of careless mistakes. We have corrected to the following.
Abstract:
Lines 32-33: This part of the final sentence is not properly constructed. Please rephrase.
Response: Thank you for your comment. We corrected the final sentence as ~based on vitamin K and exhibit the strength of biological activity can be controlled by modification of the side chain part.
Section 2
Line 80: The name of the enzyme has not been corrected. The name is vitamin K epoxide reductase, not vitamin K oxidation reductase.
Response: Thank you for your comment. We changed " vitamin K oxidation reductase " to " vitamin K epoxide reductase (VKOR)".
Lines 81-82: Understandably during revision the authors deleted “vitamin K dependent protein” and replaced by vitamin K epoxide reductase. Please correct this completely wrong sentence. Please use the plural: proteins
Response: Thank you for your comment. We changed " vitamin K epoxide reductase " to " vitamin K-dependent proteins (VKDP)".
Line 87: a reduced form
Response: Thank you for your comment. We changed " an reduced form " to " a reduced form ".
Lines 105: Factors II, VII, IX and X promote the blood coagulation, but protein C and S are inhibitors of the blood coagulation, thus they do not activate the coagulation. This part of the sentence needs to be coherently corrected.
Response: Thank you for your comment. We changed " blood coagulation factors II, VII, IX, and X and proteins C and S activate blood coagulation " to " blood coagulation factors II, VII, IX, and X activate blood coagulation".
Lines 107-108: As previously recommended to the authors, use the plural for residues, in that several Glu residues are coverted to Gla.
Response: Thank you for your comment. We changed " GGCX is an enzyme that converts glutamic acid (Glu) residue to Gla residue " to " GGCX is an enzyme that converts glutamic acid (Glu) residues to Gla residues ".
Line 109: “so that it…..” : is wrong. Please modify as follows “ so that the Gla-containing proteins can exert various ……..”
Response: Thank you for your comment. We changed " so that it " to " so that the Gla-containing proteins can exert various ".
Section 3
Line 168 “…those activity was lower “, please replace by “this activity was…..”
Response: Thank you for your comment. We changed " those activity was " to " this activity was ".
Line 181: “…..increased and higher than…….”. please add “…….increased and were higher than….”
Response: Thank you for your comment. We changed " increased and higher than " to " increased and were higher than ".
Line 250”……comound..”, please correct “compound”
Response: Thank you for your comment. We changed " comound " to " compound ".
Conclusion: Authors reintroduced the sentence including vitamins A, D and E. As I previously suggested, authors should additionally report that vitamins A, D and E, which show some structural similarities with vitamin K, present a regulatory transcriptional activity when bound to nuclear receptors, as similarly observed for vitamin K.
This is an interesting aspect which would improve the Conclusions.
Response: Thank you for your comment. We added the sentence as In addition to vitamin K, vitamins A, D, and E are all fat-soluble vitamins and have similar side chain structures. These vitamins show a similar biological activity including regulation of transcriptional activity through nuclear receptors. Such similarities are very interesting and will be important information for synthesizing new derivatives of vitamin K in the future.
Lines 279-281 The sentence “Furthermore, in order to …………..”, is overall not correctly written, and should be removed as the concept is already reported at lines 271-273.
Response: Thank you for your comment. We removed " Furthermore, in order to elucidate the action mechanism of vitamin K, to clarify the vitamin K binding protein involved in neuronal differentiation-inducing activity is needed.
The same response “We changed "The vitamin K cycle regulates the activities of both GGCX and VKOR via calumenin" to " Activity of both GGCX and VKOR are regulated by calumenin ", has been used to address two different reviewer’ comments.
Response: Thank you for your comment. We corrected the sentence according to the last reviewer’s comment.
